# Effects of *Catha Edulis* Forsk on spatial learning, memory and correlation with serum electrolytes in wild-type male white albino rats

**Abebaye Aragaw Limenie** [1]*, **Tesfaye Tolessa Dugul**[1], **Eyasu Mekonnen Eshetu** [2]

1 Department of Physiology, Addis Ababa University, College of Health Sciences, Addis Ababa, Ethiopia,
2 Department of Pharmacology, Addis Ababa University, College of Health Sciences, Addis Ababa, Ethiopia

* abebaye.aragaw@aau.edu.et

**Data Availability Statement:** All relevant data are within the paper and its Supporting information files.

## Abstract

### Background

The burdens of psychostimulant use disorders are becoming a worldwide problem. One of the psychostimulants widely consumed in Ethiopia and East African countries is *Catha edulis* Forsk (khat). However, no studies have been conducted on the cognitive effects of khat and its correlation with serum electrolytes. The present study was aimed to evaluate the effects of khat on spatial learning and memory and their correlation with serum electrolytes.

### Materials and methods

Diethyl ether and chloroform (3:1v/v ratio) were solvents used to obtain the crude khat extract in this study. T80W was used to prepare the khat juice, fresh khat leave extract. The rats were received crude khat extract subchronically (KESC) (100 mg/kg, 200 mg/kg and 300 mg/kg b.w), khat juice (KHJ 2.5 mL/kg), 2% tween 80 in distilled water (T80W- v/v, vehicle) and khat extract subacutely (KESA) (300 mg/kg). For subchronic treatment, each rat was administered for twelve weeks before Morris water maze experiment has been started, while it was administered for a week for acute treatment. Spatial learning and memory were measured using the Morris water maze model and serum sodium, calcium, potassium, and chloride were evaluated using Cobas 6000.

### Results

Spatial learning was improved with trials across the groups, while average escape latency (s) of rats received KESC 200 mg/kg (p<0.001), KESC 300 mg/kg (p<0.01) and KHJ 2.5 mL/kg (p<0.05) was significantly greater than rats that received vehicle. There was no significant difference in the latency between rats that received KESA 300mg/kg and vehicle (p>0.05). Thigmotaxis was significantly higher in rats that received all doses of khat extract (p<0.001). The time spent in the target quadrant in rats that received KESC 300 mg/kg was significantly reduced (p<0.05). Serum calcium level was inversely correlated with the escape latency (R = -0.417, p<0.05) in rats that received khat.

**Funding:** The author(s) received no specific funding for this work.

**Competing interests:** no competing interest.

## Conclusions

Khat extract and juice administered subchronically, but not subacutely, impaired learning and memory and was associated with serum calcium reduction. The neuronal basis for such alteration should be investigated.

## 1. Introduction

Substance and psychostimulants use disorders are prevalent and their burdens are getting high throughout the world [1]. Not only addiction is attributed to psychostimulants but also cognitive problems [2]. Psychostimulants such as cocaine abuse contribute to cognitive impairments [3] and a previous study showed that cognitive impairments are common in subjects with electrolyte imbalance [4].

Khat is belonging to the Celastraceae family and is an evergreen shrub [5]. It contains an amphetamine-like substance, cathinone, and is responsible for neurobehavioral alterations [6]. On the other hand, amphetamine is a widely known psychostimulant [3] and since khat contains an amphetamine-like substance [6], khat is a natural stimulant from the *Catha edulis* plant. It is widely consumed in Eastern African countries including Ethiopia [7, 8]. Khat use is associated with adverse socioeconomic and health outcomes [7–9]. Different groups of the community including students at different school levels are used to chew khat [10]. According to their response, one of the reasons for chewing is to increase their learning and memory capacity. However, some other studies found that khat reduced academic performance [11, 12], while others reported that it enhances mental alertness and performance [13, 14]. This indicates the presence of controversial findings and needs further study on the issue.

Regarding the electrolytes role in cognitive functions of the nervous system, synaptic plasticity that involves electrolytes is a primary mechanism for learning and memory. Electrolytes such as Calcium, sodium and others play important roles in neuronal functioning. Studies have found associations between low serum vitamin D levels and reduced cognitive functioning. Their homeostatic imbalance contributes to different clinical manifestations. Hyponatremia is associated with brain edema and result in different clinical manifestations that affect the cognitive functions of our nervous system [15]. Particularly, as we become aged, hyponatremia is associated with a reduction of cognitive function, pointing to hyponatremia as a risk factor of cognitive impairment [16]. On the other hand, though the link was poor, a study has suggested the link between serum parathyroid hormone, directly affects the serum calcium level, level, and cognition [17]. Vitamin D deficiencies that affect the serum calcium level are involved in the pathogenesis of various diseases that in turn become risk factors in dementia, especially Alzheimer's disease [18]. Another study also indicated that mild cognitive impairment was associated with potassium [19].

Though the correlation between the effects of khat on spatial memory and serum electrolytes has not been yet investigated, few studies have been investigated about the effects of khat on students learning capacity [11, 12, 14, 20]. The present study is, therefore, aimed to evaluate the effects of khat on cognitive functions and its correlation with serum electrolytes in wild-type male Swiss albino rats.

## 2. Materials and methods

### 2.1 Chemicals

Diethyl ether and chloroform (Siga-Aldrich, Germany), Tween 80, and 70% ethanol purchased from local suppliers in Addis Ababa, Ethiopia was used in this study.

### 2.2 Plant materials collection

Bundles of fresh khat leaves (9kg) were collected from Aweday, 525 km South-East of Addis Ababa, Ethiopia. The plant specimen was identified, and the voucher number (October 16, 2018, AA002) was given and deposited at the National Herbarium of Ethiopia, Addis Ababa University.

### 2.3 Plant material extraction

According to the previous studies [21, 22], after the edible parts of the leaves were separated and washed with tap water, the leaves were freeze-dried at -20˚C for 2 days. The leaves were crushed using mortar and pestle. Two hundred grams of freeze-dried crushed leaves were put into a conical flask wrapped with aluminum foil and a total of 400 mL of diethyl ether and chloroform (3:1v/v ratio) were added into the flask [23]. The mixture was shaken under dark conditions for 48hrs using a rotary shaker (New Brunswick Scientific Co, USA) at 120 rpm and 20˚C. Then, filtered initially using cotton gauze followed by grade I Whatman filter paper (Cat No 1001 150). The organic solvents were then removed through evaporation using Rota-vapor under a controlled temperature of 36˚C, rotation of 120 rev/min, and 240 Pascal negative pressure. The water in the extract was removed through lyophilization and the dry residue was weighed and stored in a desiccator till used.

The khat juice (KHJ) was prepared from 12gm of fresh khat leaves for every 1000 gm animal body weight. Thus, for a rat weighing about 300gm, the amount of fresh khat leaves required is 3.6 gm. The fresh leaves in 2% tween 80 in distilled water (T80W) were crushed using a blender machine. The juice was then squeezed and filtered using the gauze and grade I Whatman filter paper (Cat No 1001 150). The amount of T80W in distilled water used to extract the given weight of leaves was determined based on the total weight of each rat and standard vehicle volume (2.5 mL/kg b.w).

### 2.4 Animal preparation

A total of 36 adult wild-type male Swiss albino rats aged between 7–8 weeks weighing between 213 and 229g were used. The rats were purchased from a Laboratory Animal Breeding Section of the Ethiopian Public Health Institution. Three rats per plastic cage under natural light and dark (12:12hrs) cycles at room temperature were housed. Water and standard pellet diet were available *ad libitum* throughout the study period. Rats were weighed twice a week to ensure appropriate dosing. All the studies were conducted under the guidelines for animal research as detailed in the NAP guidelines for Care and Use of Laboratory Animals [24]. The study was approved by the Institutional Review Board ethical committee in Addis Ababa University.

### 2.5 Grouping and dosing

Rats were randomly assigned into six groups (n = 6 /group) and received T80W, khat extract subchronically (KESC) (100 mg/kg, 200 mg/kg and 300 mg/kg), khat juice (KHJ 2.5 mL/kg), and khat extract subacutely (KESA) 300 mg/kg. The subchronic groups of rats were received the test substances and vehicle for twelve weeks before the behavioral experiment and during the Morris water maze experiment, while it was for one week in the case of the acute group.

Rats that received the T80W (vehicle) were taken as controls. The doses for the extract were selected based on the previous report [25].

## 2.6 Preparation of test substances and volume determination

Fresh solutions of extract, KHJ, and vehicle were prepared every day. The khat extract was dissolved in T80W. The dose of the extract administered in each rat was calculated from the selected doses (100 mg/kg, 200 mg/kg, and 300 mg/kg) and the total b.w of each rat. For instance, for a rat in the lower dose group (100 mg/kg) weighing for 300gm, the dose of the extract administered was 30mg (300mg*100mg/1000gm). The appropriate standard vehicle volume (2.5 mL/kg b.w) was used to determine how much volume was used to dissolve the calculated dose of khat extract. Each rat in its respective group received a single daily oral administration of vehicle, khat extract, KHJ. The final volume for each rat was 1mL and all substances were administered orally using metal gavage.

## 2.7 Morris water maze performance test

**2.7.1 Apparatus and experimental procedure.** The water maze consisted of a uniformly blue circular pool (180cm in diameter and 52 cm high) filled to a depth of 25 cm with water (24±2˚C) [23]. Water was heated using an electric heater every morning before the experiment. The maze was divided into North (N), South (S), East (E) and West (W) equal quadrants. The immovable platform (10x24cm) where each rat can escape from the water and external fixed position maze cues for each rat to navigate the hidden platform during the acquisition phase were used.

The previous procedure used by Frick et al. [26] and Kimani et al. [27] was applied. Briefly, one day after habituation, acquisition trials were started. The platform was placed in the middle of the South-East (SE) quadrant. It was 1cm above the water surface on the first day of acquisition trials and then I cm below the water surface on the remaining trials. The position of the platform was fixed throughout the acquisition trials. 24hrs after test substances treatment, each rat was placed into the maze facing the wall of the pool at one of the randomly selected starting positions. Different starting positions were used for different trials for each rat. Each rat was allowed to swim for 120s to find the hidden platform. If a rat failed to get the hidden platform within this time limit, the rat was manually guided into the platform and 120s was taken as the escape latency. The rats were allowed to last on the platform for 30s before being returned into the cage. The acquisition trials were conducted for 5 consecutive days with 4 trials per/day having an intertrial interval of 10 min. The performances of each rat were videotaped by the camera mounted 1 m above the maze to be replayed later into a computer. Escape latency (s), swim path-length (cm), swim speed (cm/s) and thigmotaxis (s) were measured during each trial and the average value was taken for each acquisition day.

The probe trial was conducted 24hrs after the last acquisition trial. The platform was removed and each rat was allowed to swim for 120 s starting from North-West (NW). Time spent in each quadrant (s), transfer latency (s), swim path-length (cm), and speed (cm/s) in the target quadrant (Tq), and the number of crossovers into the Tq were measured.

## 2.8 Serum electrolyte determination

At the end of the neurobehavioral study and 24 hrs after the last test substances administration, whole blood was collected from each rat for serum electrolytes assay. The procedure used by Alsalahi *et al.* [28] was applied during serum electrolyte determination. Five milliliters of fasting blood from each rat were collected into plain tubes through cardiac puncture. The cardiac puncture was conducted after each rat was anesthetized using sodium pentobarbital (1ml/kg

rat b.w). Then, the tubes were allowed to stand to clot. Following clotting, the tubes were centrifuged immediately with a speed of 6000rpm for 10 minutes to obtain the supernatant. The supernatant was slowly transferred into a new Eppendorf tube for sodium ($Na^+$), potassium ($K^+$), calcium ($Ca^{2+}$), and chloride ($Cl^-$) analysis using an electrolyte machine (Automated electrolyte Analyzer: Cobas- 6000).

## 2.9 Statistical analysis

SPSS version 21.0 and Microsoft Excel were used during data analysis. The values were expressed as mean ±S.E.M. Differences in mean values between groups were analyzed using one-way ANOVA followed by post hoc analysis. Two ways repeated measures of ANOVA followed by Bonferroni test, t-test, and Pearson's correlation were also used in this study.

# 3. Results

## 3.1 Escape latency during acquisition trials

The latency was significantly reduced across trials among the groups ($F_{(1.24, 6.20)} = 45.70$, $p<0.001$). Treatment and its interaction with trails had also significant effect on the latency ($F_{(6, 30)} = 14.16$, $p <0.001$ and $F_{(24,120)} = 5.75$, $p<0.001$, respectively. Bonferroni estimated marginal test didn't show a significant difference in latency between groups (Fig 1). However, Tukey's test analysis showed that the average latency in rats that received KESC 200 mg/kg ($p<0.001$), KESC 300 mg/kg ($p<0.01$), and KHJ 2.5 mL/kg ($p<0.05$) was significantly higher than in rats that received T80W (Table 1). The latency of rats that received KESA 300 mg/kg was significantly reduced compared with rats that received KESC300 mg/kg (15.40±.75 vs 30.83±1.54, $p<0.01$).

## 3.2 Swim path-length during acquisition trials

The path-length was significantly reduced with trials among the groups ($F_{(1.19, 5.94)} = 29.79$, $p<0.01$). Treatment and its interaction with trials had also a significant effect on the path-length ($F_{(6, 30)} = 14.67$, $P<0.001$ and $F_{(24,120)} = 11.32$, $p<0.001$, respectively). The Bonferroni estimated marginal mean analysis indicated that there was a significant change in swim path-length among rats that received KHJ 2.5 mL/kg subchronically ($p<0.01$) and KESA 300 mg/kg ($p<0.05$) compared with rats that received T80W (Fig 2). Rats that received KESA 300 mg/kg

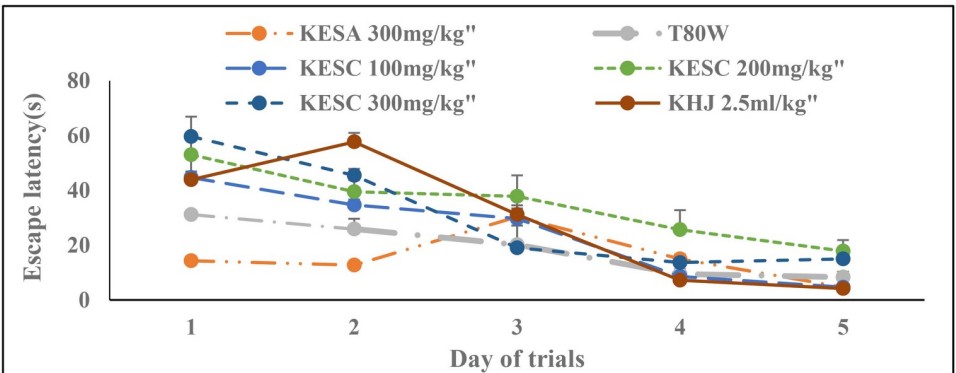

**Fig 1. Effect of khat extract and juice on escape latency during acquisition trials.** Each point across the line represents the mean ± SEM of escape latency in rats (n = 6 in each group) that received T80W, KESC (100 mg/kg, 200 mg/kg and 300 mg/kg), KESA 300 mg/kg and KHJ 2.5 mL/kg. Each group of rats was compared with rats that received T80W. KESC: khat extract subchronic, KESA: khat extract subacute, KHJ: khat juice.

**Table 1. Effects of khat on average escape latency, swim path-length, swimming speed and thigmotaxis in rats.**

| Group | Parameters | | | |
|---|---|---|---|---|
| | Escape latency(s) (M±SEM) | Swim Path length(cm) (M±SEM) | Swim speed(cm/s) (M±SEM) | Thigmotaxis (s) (M±SEM) |
| **T80W** | 19.00±2.31 | 416.67±59.37 | 21.83±1.79 | 9.67±.1.17 |
| **KESA 300 mg/kg** | 15.17±.75 | 703.17±38.65** | 46.48±1.89*** | 6.50±34*** |
| **KESC 100 mg/kg** | 24.50±1.09 | 801.33±41.23*** | 32.97±1.99*** | 18.83±.79*** |
| **KESC 200 mg/kg** | 34.83±2.46*** | 841.67±63.61*** | 24.19±1.03 | 23.83±2.01*** |
| **KESC 300 mg/kg** | 30.83±1.54** | 822.50±35.48*** | 26.78±.72 | 21.67±1.20*** |
| **KHJ 2.5 mL/kg** | 28.83±2.63* | 775.17±58.97*** | 27.13±1.03 | 20.50±1.57*** |

Each point represents the mean ± SEM of escape latency, swim path-length, swim speed and thigmotaxis in rats (n = 6/ group) that received T80W, KESC (100 mg/kg, 200 mg/kg and 300 mg/kg), KESA (300 mg/kg) and KHJ 2.5 mL/kg.

***$p < 0.001$, **$p < 0.01$ and *$p < 0.05$ when each group of rats was compared with rats that received T80W.

KESC: khat extract subchronic, KESA: khat extract subacute, KHJ: khat juice.

($p<0.01$), KESC 100 mg/mg ($p<0.001$), KESC 200 mg/kg ($p<0.001$), KESC 300 mg/kg ($p<0.001$) and KHJ 2.5 mL/kg ($p<0.001$) swam significantly more path-length than rats that received T80W (Table 1).

### 3.3 Swimming speed (cm/s) during acquisition trials

Significant change was observed in the swim speed across trials ($F_{(4, 20)} = 14.63$, $p < 0.001$). The treatment and its interaction with trials had also a significant effect on swim speed ($F_{(2.38, 11.92)} = 23.93$, $p<0.001$ and $F_{(3.01, 15.06)} = 4.87$, $P<0.05$, respectively). The Bonferroni estimated marginal mean test indicated that there was no significant difference in swim speed between groups (Fig 3). However, rats that received KESC 100 mg/kg had significantly greater swim seed than rats that received T80W ($p<0.001$) and KESC 200 mg/kg ($p<0.05$). Rats that received KESA 300 mg/kg swam significantly faster than rats that received T80W ($p <0.001$), and KESC 300 mg/kg ($p <0.001$).

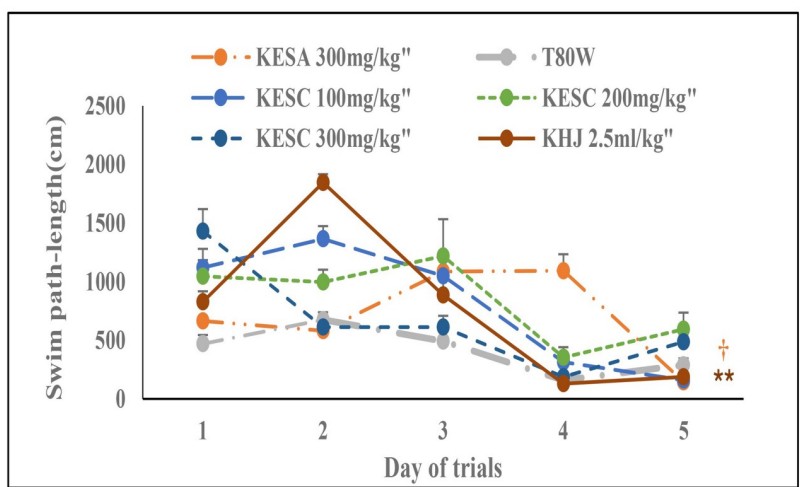

**Fig 2. Effect of khat extract and Juice on swim path-length during acquisition trials.** Each point across the line represents the mean ± SEM of swim path-length in rats (n = 6 / group) that received T80W, KESC (100 mg/kg, 200 mg/kg and 300 mg/kg), KESA (300 mg/kg) and KHJ 2.5 mL/kg. KESC: khat extract administered subchronically, KESA: khat extract subacute.

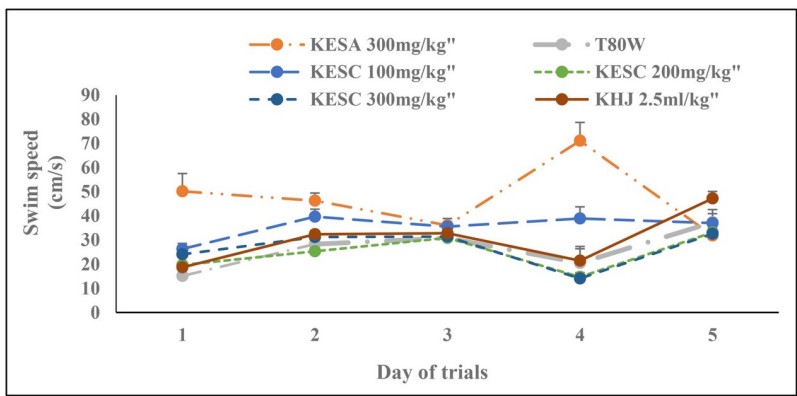

**Fig 3. Effect of khat extract and juice on swim speed during acquisition trials.** Each point across the line represents the mean ± SEM of swimming speed in rats (n = 6 / group) that received T80W, KESC (100 mg/kg, 200 mg/kg and 300 mg/kg), KESA (300 mg/kg) and KHJ 2.5 mL/kg. Each group of rats was compared with rats that received T80W. KESC: khat extract subchronic, KESA: khat extract subacute, KHJ: khat juice.

### 3.4 Thigmotaxis during acquisition trials

The time spent at the periphery of the maze was significantly reduced across trials ($F_{(4, 20)}$ = 55.30, P<0.001). The treatment and its interaction with trials had also a significant effect on thigmotaxis ($F_{(1.98, 9.89)}$ = 23.96, p<0.001 and $F_{(2.49, 12.45)}$ = 5.06, p<0.05, respectively). The Bonferroni estimated marginal mean test showed that rats received KESC 100 mg/kg (p<0.01), KESC 200 mg/kg (p<0.01) and KESC 300 mg/kg (p<0.01) had significantly higher thigmotaxis than rats that received T80W (Fig 4).

Rats that received KESC100 mg/kg (p<0.001), KESC 200 mg/kg (p<0.001), KESC 300 mg/kg (p<0.001) and KHJ 2.5 mL/kg (p<0.001) had also significantly greater thigmotaxis than rats that received T80W (Table 1). However, increased in thigmotaxis among these groups, except rats that received KHJ 2.5 mg/kg (R = .937, p <0.01), of rats was not significantly correlated with the escape latency (R = .677, p>0.05; R = .750, p>0.05 and R = .752, p>0.05, respectively, Fig 5). The thigmotaxis in rats that received KESA 300mg/kg was significantly less than in rats that received KESC 300 mg/kg (p<0.001) and vehicle (p<0.001).

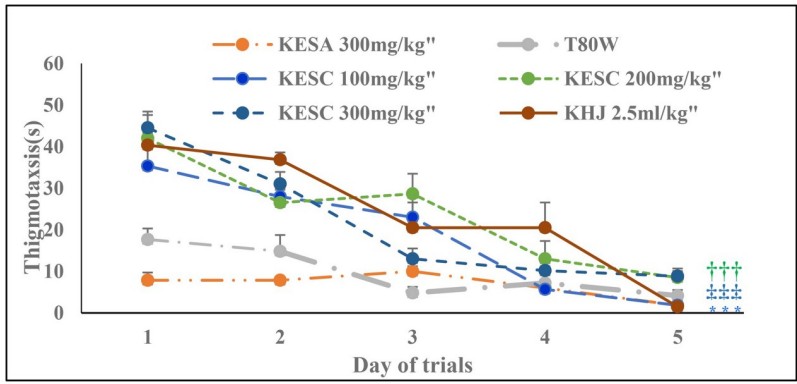

**Fig 4. Effect of khat extract and juice on thigmotaxis during acquisition trials.** Each point across the line represents the mean ± SEM of thigmotaxis in rats (n = 6 / group) that received T80W, KESC (100 mg/kg, 200 mg/kg and 300 mg/kg), KESA (300 mg/kg) and KHJ 2.5 mL/kg. KESC: khat extract administered subchronically, KESA: khat extract subacute, khat juice.

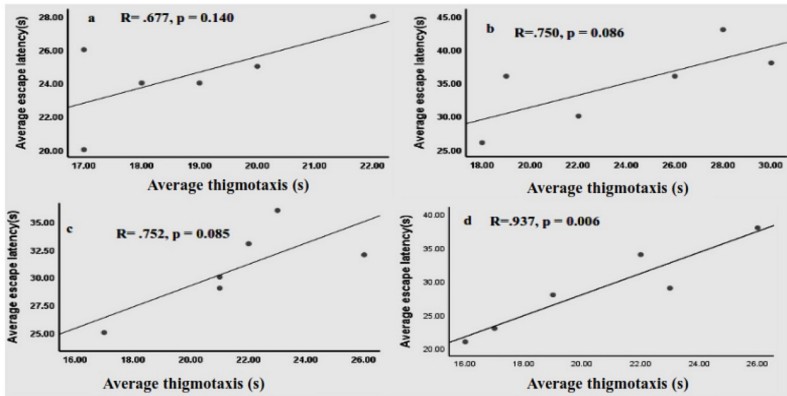

**Fig 5. Correlation between average escape latency and thigmotaxis of rats (n = 6 / group) that received KESC 100 mg/kg (a), KESC 200 mg/kg (b), KESC 300 mg/kg (c) and KHJ 2.5 mL/kg (d).** R: correlation coefficient.

### 3.5 Time spent in the target quadrant during probe trials

The paired t-test indicated that rats that received KESC 300 mg/kg had significantly less time spent in the target quadrant (Tqt) than in the left adjacent (t $_{(5)}$ = -3.75, p<0.05) and opposite quadrant (t $_{(5)}$ = -2.598, p<0.05). However, the rats that received KESA 300 mg/kg had significantly higher Tqt than in left adjacent (t $_{(5)}$ = 15.62, P<0.001), right adjacent (t $_{(5)}$ = 15.09, p<0.001) and opposite (t $_{(5)}$ = 12.72, p<0.001). Rats that received T80W had significantly higher Tqt than in the opposite quadrant (t $_{(5)}$ = 2.68, p<0.05). The Tqt by rats that received KHJ 2.5 mL/kg was significantly less than in the left adjacent quadrant (t $_{(5)}$ = -3.75, p<0.05, Table 2).

Independent t-test also revealed that the Tqt (s) in rats that received KESC 300 mg/kg was significantly reduced compared with rats that received T80W (, p<0.05, Table 3).

### 3.6 Transfer latency into the target quadrant during probe trials

There was a significant difference in the first target site crossover among the groups (F $_{(6, 35)}$ = 35.23, p < 0.00). The post hoc Tukey's test analysis indicated that the time was taken by rats that received KESC 100 mg/kg, KESC 200 mg/kg, KESC 300 mg and KHJ 2.5mL before they

**Table 2. Effects of khat on time spent(s) in target and other quadrants during the probe trials.**

| Group | Time spent(s) in different quadrants | | | | |
|---|---|---|---|---|---|
|  | SE(Tq) | SW | NE | NW | Probe trial conducted |
| **T80W** | 31.33±2.16 | 28.12±1.03 | 25.11±1.42 | 25.03±.55* | 24hrs after the last acquisition trial |
| **KESA 300 mg/kg** | 36.50±.89 | 26.00±.78** | 24.17±.55*** | 23.33±.72*** | |
| **KESC 100 mg/kg** | 27.83±1.01 | 28.77±.88 | 26.96±.86 | 26.61±.53 | |
| **KESC 200 mg/kg** | 30.83±3.31 | 27.77±1.61 | 23.34±1.46 | 26.06±1.35 | |
| **KESC 300 mg/kg** | 24.83±.98 | 29.28±.56* | 28.55±.87 | 27.34±.15* | |
| **KHJ 2.5mL** | 25.00±.97 | 29.52±.59* | 27.99±.76 | 27.52±1.21 | |

Each point represents mean ± SEM of time spent (s) in different quadrants in rats (n = 6 in each group) that received T80W, KESA (300 mg/kg), KESC (100 mg/kg, 200 mg/kg and 300 mg/kg) and KHJ 2.5 mL/kg.

***p <0.001, **p< 0.01 and *p < 0.05 when the time spent in the Tq (SE) was compared with the time spent in left adjacent (SW), right adjacent (NE) and opposite (NW) quadrants.

KESC: khat extract subchronic, KESA: khat extract subacute, KHJ: khat juice and Tq: target quadrant.

**Table 3. Effects of khat on time spent (s) in Tq during probe trials.**

| Group | KESA300 mg/kg | KESC100 mg/kg | KESC200 mg/kg | KESC 300 mg/kg | KHJ 2.5 mL/kg |
|---|---|---|---|---|---|
| **Tqt(s)** | 36.5±0.89 | 27.50±1.06 | 26.17±2.24 | 23.83±1.22* | 25.00±.97 |
| **T80W** | 31.33±2.16 | | | | |

Each point represents the mean ± SEM of time spent (s) in different quadrants in rats (n = 6 in each group) that received T80W, KESA (300 mg/kg), KESC (100 mg/kg, 200 mg/kg and 300 mg/kg) and KHJ 2.5 mL/kg.

***p <0.001, **p< 0.01 and *p < 0.05 when Tqt (SE) in each group was compared with rats that received T80W.

KESC: khat extract subchronic, KESA: khat extract subacute, KHJ: khat juice. Tqt(s): time spent (s) in the target quadrant.

entered into the target quadrant (Tq) was significantly greater compared to the control (7.33±0.84 vs 3.50±0.50, p<0.05; 8.67±0.56 vs 3.50±0.50, p<0.01; 9.50±0.56 vs 3.50±0.50, p<0.001 and 16.17±1.11, p<0.001, respectively) in a dose-dependent manner (Fig 6).

### 3.7 Effects of khat on serum electrolytes

Analysis of one-way ANOVA indicated that significant differences were observed in the concentrations of $Na^+$ ($F_{(5,30)}$ = 12.17, P<0.001), $K^+$ ($F_{(5,30)}$ = 2.99, p < 0.05), $Ca^{++}$ ($F_{(5,30)}$ = 3.32, p<0.05) and as well as $Na^+ / K^+$ ratio ($F_{(5,30)}$ = 5.54, p<0.01) whereas there was no significant difference in $Cl^-$ ($F$ (5,30) = 0.78, p>0.05) concentration. The post hoc analysis revealed that the mean serum $Na^+$ concentrations in rats that received KESC 100 mg/kg (p<0.01), KESC 200 mg/kg (p<0.001), KESC 300 mg/kg (p<0.001) and KHJ 2.5 mL/kg (p<0.001) were significantly less than in rats that received T80W (Table 4). Rats that received KESC 200 mg/ kg (p<0.05) and 300 mg/kg (P<0.05) had significantly reduced $Ca^{++}$ concentration. The concentration of $K^+$ in rats that received KESC 300 mg/kg was insignificantly increased compared with rats that received T80W (Table 4) and significantly higher than rats that received KESC 100 mg/kg (p<0.05) while $Na^+/K^+$ ratio was significantly reduced (p<0.001).

Although a significant correlation between the escape latency (s) and mean serum $Na^+$ (R = -0.383, p > 0.05) and $Na^+/K^+$ ratio (R = -0.365, p > 0.05) level was not observed it was inversely correlated with serum $Ca^{++}$ level in rats that received khat (Fig 7a and 7b). However,

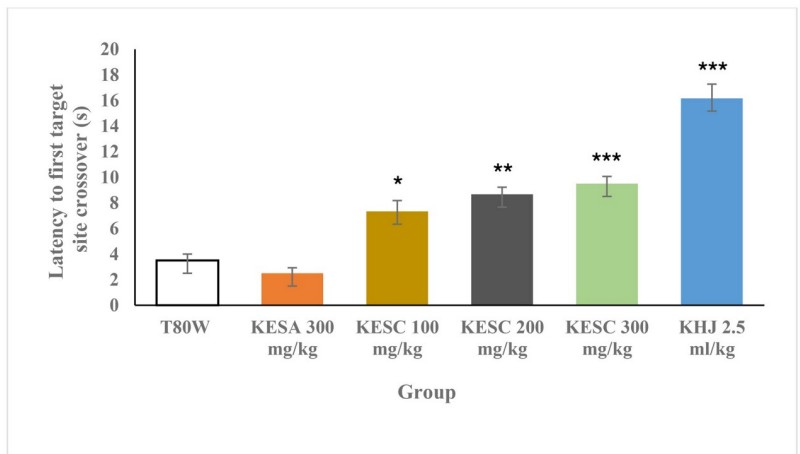

**Fig 6. Latency to the Tq during the probe trial.** Each bar represents the mean ± SEM of the transfer latency to the Tq in rats (n = 6 in each group) that received T80W, KESA (100 mg/kg, 200 mg/kg, 300 mg/kg), KESC (300 mg/kg) and KHJ 2.5 mL/kg. ***P < 0.001, **P < 0.01 and *P < 0.05 when each group of rats was compared with rats that received T80W. KESC: khat extract subchronic, KESA: khat extract subacute, KHJ: khat juice.

**Table 4. Effects of khat on serum electrolytes.**

| Group | Electrolytes (mmol/L), M±SEM | | | | |
|---|---|---|---|---|---|
| | Na+ | K+ | Ca++ | Cl⁻ | Na⁺/K⁺ |
| T80W | 145.67±1.31 | 5.88±.44 | 2.78±.23 | 101.59±.77 | 25.55±2.06 |
| KESC 100 mg/kg | 140.50±.99** | 4.28±.45 | 2.37±.04 | 102.43±.95 | 34.29±2.89 |
| KESC 200 mg/kg | 136.67±.92*** | 6.74±1.43 | 2.08±.19* | 99.95±.71 | 23.28±2.94 |
| KESC 300 mg/kg | 137.17±.79*** | 9.14±1.22 | 2.00±.19* | 100.26±.68 | 16.44±2.26 |
| KHJ 2.5 mL/kg | 138.67±.61*** | 6.95±.51 | 2.43±.05 | 100.76±1.22 | 20.49±1.47 |

Each point represents the mean ± SEM of electrolytes in rats (n = 6/group) that received T80W, khat extract subchronically (KESC) (100 mg/kg, 200 mg/kg and 300 mg/kg) and KHJ 2.5 mL/kg.

***P <0.001, **P <0.01 and *P <0.05 when each group of rats was compared with rats that received T80W.

T80W: tween 80 in distilled water, KESC: khat extract subchronic and KHJ: khat juice.

there was no significant correlation between the escape latency and $Ca^{++}$ (R = -0.322, p>0.05) in rats that received T80W observed (Fig 7).

## 4. Discussion

### 4.1 Escape latency during acquisition trials

Extract at the middle and higher doses and khat juice given subchronically increased the average escape latency, revealing that the crude khat extract and juice affect learning negatively. Kimani and Nyongesa [21] also indicated that the middle and the higher doses of khat extract impaired learning. However, the study conducted by Mohammed *et al*. [8] indicated that latency was increased significantly across trials in rats that received the lower dose of the extract. The difference in this finding could be attributed to the test protocol and animal model used to evaluate the effect of khat on spatial learning. In our study, khat extract and juice were administered before and during behavioral experiments. However, it was only before the experiment in the study conducted by Mohammed *et al*. [8].

Latency in rats that received khat extract subacutely was significantly reduced compared with rats that received the same dose administered subchronically. However, a significant difference was not observed in the latency when this parameter in the rat that received the extract subacutely compared with rats that received the vehicle. This outcome revealed that subacute administration of khat extract didn't affect spatial learning. Nevertheless, amphetamine, a chemical with the same structure and function as cathinone in khat, showed spatial learning impairment administered subacutely as compared with control [29].

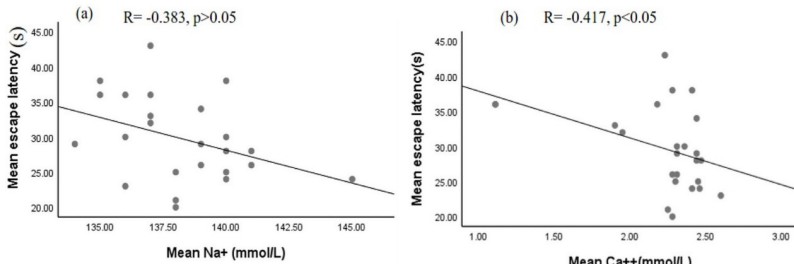

**Fig 7. Correlation between escape latency and serum Na⁺ level (a) and Ca²⁺ (b) in rats that received khat extract and juice.** R: correlation coefficient.

## 4.2 Swim path-length and speed during acquisition trials

The swim path-length was reduced across trials and it was more pronounced in rats that received khat extract subacutely indicating that rats administered with khat extract subacutely found the hidden platform before traveling more distance in the maze. However, the average swim-path length moved by the rats that received all doses of khat extract and juice subchronically was significantly greater than the rats that received the vehicle. Previous studies indicated that rats that received cocaine [3] and amphetamine [29, 30] swam longer path-length before they found the hidden platform when compared with control. Rats that received the lower dose of extract subchronically had a significantly greater speed than rats that received the middle dose of the extract and vehicle. This indicates that the different doses of the khat extract could have different effects on the brain neurotransmitters. According to other studies, the lower and higher doses of amphetamine induced a differential response on the level of brain neurotransmitters [31] and its locomotor activity was maximum at the lower dose [32].

On the other hand, rats that received the higher dose of extract subacutely had significantly greater swimming speed than rats that received the same dose of the extract administered subchronically. This indicated that the locomotor effect of the khat was stronger at a lower dose and administered for a short time. A similar finding was reported by Kimani and Nyongesa [21] in that the average swimming speed was significantly higher in mice that received the lower dose compared with mice that received the higher and middle doses of extract and vehicle.

## 4.3 Thigmotaxis during acquisition trials

Rats at all dose groups of the extract and khat juice administered subchronically had significantly higher thigmotaxis than rats received the vehicle and extract subacutely. Yet, elevated stress and anxiety increase thigmotaxis [3], and stress affects learning and memory [33], a significant correlation between thigmotaxis and escape latency was not observed in rats that received khat extract. Though stress affects either the short-term or long-term memory based on the duration and, the intensity of the stress [34, 35], Mendez *et al.* [3] indicated that though thigmotaxis was significantly higher in rats administered with cocaine, there was no significant correlation between latency and thigmotaxis. However, a significant correlation was observed between thigmotaxis and latency in rats that received khat juice. The differential effects of khat on thigmotaxis and the escape latency could be attributed to the effect of extraction processes on the phytochemical constituents in khat [36].

These findings showed that khat extract administered in rats for a longer period had an anxiogenic effect, and thigmotaxis in rats that received khat juice had effects on time taking to find the hidden platform during the acquisition trials in rats that received khat juice. However, subacute administration of khat extract didn't contribute to stress, thigmotaxis, and reduction of learning and memory. Similarly, a previous study reported that acute administration of khat in an animal model reduced stress levels and showed antidepressant-like activities [23].

## 4.4 Time spent in the target quadrant during probe trials

Rats administered with a higher dose of khat extract and khat juice subchronically had less time spent in the target quadrant than the time spent in other quadrants. However, like the previous study [6], rats that received the same dose of khat extract subacutely spent more time in the target quadrant in our study. These findings revealed that spatial memory was impaired at the higher dose and prolonged administration of khat extract and juice. However, a previous study [21] indicated that the higher dose (360mg/kg b.w) improved memory but it was impaired by the lower (40mg/kg b.w) and middle (120mg/kg b.w) doses of khat extract. The disagreement between these findings could be attributed to the extraction process, route of

administration, type of animal model used in the study, probe trial time, and time of behavioral test conducted after khat administration.

## 4.5 Transfer latency into the target quadrant during the probe trial

The time taken before the initial entry into the target quadrant by rats administered with all doses of khat extract and juice subchronically was significantly greater than in rats administered with the vehicle. This indicated that rats administered with khat took more time before they entered into the target quadrant.

The impairment in the Morris water maze task performance in this study could be attributed to an alteration in brain structures, brain glucose metabolism, sleep, serum electrolytes, antioxidants, and oxidative stress, hemoglobin, and neurotransmitter levels. A study conducted before indicated that amphetamine destroyed hippocampal cells, disrupted the blood-brain barrier, and increased reactive oxygen specious in brain areas involved in Morris water maze performance [37]. Other studies showed that khat impaired mood [38], sleep [39], working memory, and inhibitory control response [11]. Synthetic cathinone arrested hippocampal cell growth [40] and khat extract destroyed the prefrontal cortex [41]. Thanos *et al.* [42] showed that amphetamine enlarged brain striatal volumes and reduced glucose metabolism.

## 4.6 Effects of khat on serum electrolytes and their correlation with learning

Like a study conducted by Mwaniki *et al.* [43], all doses of khat extract and juice in this study reduced serum sodium levels significantly compared to vehicle. Amphetamine, cathinone in khat like substance, also reduced the serum sodium level [44, 45]. In this study, calcium was also significantly reduced in rats that received the middle and higher doses of khat extract. Reduced calcium concentration was reported in subjects with depression [46] and depression was more common subjects administered with khat [10, 47]. However, another study indicated that the serum calcium level was significantly increased among normal subjects and diabetic patients who chewed khat [48].

In this study, a significant inverse correlation between escape latency and calcium was observed in rats that received khat extract, indicating that the less the serum calcium level was the higher the escape latency. The more the escape latency signifies a reduction in memory and the spatial memory deficits in this study could be attributed to the serum calcium level effect of khat extract.

On the other hand, according to the previous studies, depression is associated with the serum calcium level [46], and anxiety showed an effect on memory [49, 50]. Although a significant correlation between serum sodium and escape latency was not observed in our study, previous reports showed that reduced serum sodium levels contributed to cerebral edema and cognitive problems [4, 51]. Studies also showed that blood pressure which is associated with cognitive impairments [52, 53] was affected by serum electrolytes imbalance [54].

In conclusion, khat extract at the higher dose and prolonged administration impaired spatial learning and memory. However, khat extract administered for a short period couldn't affect these cognitive functions. Effects of khat on neurochemical, structural, molecular, and neurotransmitter signaling pathways for neurobehavioral findings observed in this study remain to be established.

## Supporting information

**S1 Abstract.**
(DOCX)

**S1 Data.**
(XLSX)

**S1 File.**
(DOCX)

## Acknowledgments

We would like to express our appreciation to Dr. Nigusie Deyessa for his support during statistical analysis. We are indebted to Demere Bekilla and other staff in the Department of clinical chemistry, Ethiopian public health institute for their assistance during electrolyte analysis. We would like also to express our indebtedness to Tesfaye Getachew for his technical support during blood sample collection during serum electrolyte analysis.

## Author Contributions

**Conceptualization:** Abebaye Aragaw Limenie.

**Data curation:** Abebaye Aragaw Limenie.

**Formal analysis:** Abebaye Aragaw Limenie.

**Investigation:** Abebaye Aragaw Limenie.

**Methodology:** Abebaye Aragaw Limenie, Eyasu Mekonnen Eshetu.

**Project administration:** Tesfaye Tolessa Dugul.

**Resources:** Tesfaye Tolessa Dugul.

**Supervision:** Eyasu Mekonnen Eshetu.

**Visualization:** Tesfaye Tolessa Dugul.

**Writing – original draft:** Abebaye Aragaw Limenie.

**Writing – review & editing:** Abebaye Aragaw Limenie, Tesfaye Tolessa Dugul, Eyasu Mekonnen Eshetu.

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
