## [Decision Letter · Decision Letter 0]

23 Nov 2021

PONE-D-21-29239Title-Effects of Catha Edulis Forsk on Spatial Cognition and correlation with serum electrolytes in Wild- type Male White Albino RatsPLOS ONE

Dear Dr. Limenie,

Thank you for submitting your manuscript to PLOS ONE. After careful consideration, we feel that it has merit but does not fully meet PLOS ONE’s publication criteria as it currently stands. Therefore, we invite you to submit a revised version of the manuscript that addresses the points raised during the review process.

We look forward to receiving your revised manuscript.

Kind regards,

Michelle Melgarejo da Rosa

Academic Editor

PLOS ONE

Journal Requirements:

“This research was supported by Addis Ababa University and the Department of Physiology.”

“The author(s) received no specific funding for this work”

Reviewers' comments:

Reviewer's Responses to Questions

**Comments to the Author**

1. Is the manuscript technically sound, and do the data support the conclusions?

Reviewer #1: Yes

Reviewer #2: Yes

2. Has the statistical analysis been performed appropriately and rigorously? 

Reviewer #1: Yes

Reviewer #2: Yes

3. Have the authors made all data underlying the findings in their manuscript fully available?

Reviewer #1: Yes

Reviewer #2: Yes

4. Is the manuscript presented in an intelligible fashion and written in standard English?

Reviewer #1: No

Reviewer #2: Yes

5. Review Comments to the Author

Reviewer #1: GENERAL COMMENTS

- Please carefully revise language throughout the manuscript.

TITLE

- Correct to “Catha edulis Forsk” (Forsk should not be in italics)

ABSTRACT

- Line 2: “Forsk” should not be in italics

- Lines 4-5. The sentence indicating the objectives (“The present study was aimed to evaluate the effects of khat on cognitive functions and its correlation with serum electrolytes”) can be revised to be more specific about the cognitive functions evaluated. As it stands, it is almost a repeat of the previous sentence in the Abstract.

- Line 7. The information about the total number and weight of the animals can be removed of the Abstract. On the other hand, I consider important mentioning the solvent in which the crude khat extract was obtained and what is “khat juice”.

- Make clearer how long “subchronical” and “subacute” treatments last.

- Please mention what electrolytes were evaluated.

- The sentence “The data were analyzed using SPSS version 21.0 and Microsoft Excel” should be removed.

- I suggest putting sample/treatments abbreviations in capital letters: KESC, KESA, KHJ (here in Abstract and along all the manuscript)

INTRODUCTION

- How is khat popularly used as a psychostimulant?

- Have the authors tried to investigate the presence of Cathinone in the samples?

- The role of serum electrolytes on cognitive functions should be more explored in introduction.

- Please be clearer in the sentence: “Regardless of its adverse effect (8 and 9), misperceptions of its learning and memory effects are reflected among students”.

MATERIALS AND METHODS

- Section 2.3. Please revise the sentence “The khat juice (khJ) was prepared from 12g/kg body weight (b.w) of fresh leaves”. I did not understand it.

- Please also revise “The amount of T80W in distilled water used to extract the given weight of leaves…”

- Section 2.4. Is there an approval number/protocol by the ethics committee?

- Section 2.5. What really means “during the experiment”?

- Revise the sentence “The dose of the extract administered in each rat was calculated from the selected doses”

RESULTS

- The lack of analysis of the chemical composition of the samples is an important negative point. Authors should try to characterize the extract and juice evaluated.

- Quality of the images seem bad in the version sent for review.

- Many of the figures can be merged in one.

DISCUSSION

- Please avoid extensive repeat of the results.

- Again, the lack of chemical analysis makes the discussion little in-depth and very descriptive.

- There are several comparisons with other studies performed with khat. However, it should be clarified the samples evaluated in all these studies. Extracts? What types? Compounds?

Reviewer #2: The subject of the manuscript is interesting. The authors focused one of the psychostimulants widely consumed in Ethiopia and East African countries - Catha edulis Forsk (khat).

The novelty of the present manuscript results from the fact that no studies have been conducted on the cognitive effects of khat and its correlation with serum electrolytes.

The authors evaluated the effects of Catha edulis on cognitive functions and its correlation with serum electrolytes.

The text is clear and easy to read.

The design research is appropriate.

The results are consistent and clearly presented.

The presented data are supported by the conclusions.

The reference list is variously.

At the reference list the name of species Catha edulis is not written with italic (example: reference numbers 6, 7, 9, 12, 16, 17, 20, 22, 23, 38, 42, 43). Please to correct!

6. PLOS authors have the option to publish the peer review history of their article (what does this mean?). If published, this will include your full peer review and any attached files.

Reviewer #1: No

Reviewer #2: No

---

## [Author Response · Author response to Decision Letter 0]

8 Jan 2022

We authors would like to acknowledgement the reviewers for their critical looking at the article and constructive comments for possible publication. We also appreciate the editor for your great contribution and considered our manuscript entitled “Effects of Catha Edulis Forsk on Spatial Learning, Memory and Correlation with Serum Electrolytes in Wild- Type Male White Albino Rats” (PONE-D-21-29239R1) for possible publication if the comments are well addressed and satisfied the editors and reviewers. We have carefully reviewed the comments and have revised the manuscript and abstract accordingly. Our responses are given in a point-by-point manner in the table uploaded. We tried to address all the specific comments from the editor and reviewers. We hope the revised version is now suitable for publication and look forward to hear from you in due course. If you need further information about it, we are waiting for you.

Sincerely

Abebaye Aragaw Limenie

Department of Physiology, Addis Ababa University, College of Health Sciences, Addis Ababa, Ethiopia P.O. Box 9086, email, abebaye.aragaw@aau.edu.et

On the behalf of co-authors

---

## [Editor Report · Decision Letter 1]

12 Jan 2022

Effects of Catha Edulis Forsk on Spatial L earning, Memory and Correlation with Serum Electrolytes in Wild- Type Male White Albino Rats

PONE-D-21-29239R1

Dear Dr. Limenie,

We’re pleased to inform you that your manuscript has been judged scientifically suitable for publication and will be formally accepted for publication once it meets all outstanding technical requirements.

Kind regards,

Michelle Melgarejo da Rosa

Academic Editor

PLOS ONE

Additional Editor Comments (optional):

Dear Authors, We appreciate the effort in following the suggestions from reviewers. After the final revision, we are glad to inform that the present manuscript is accept from Plos One Journal. Best Regards.
---

## [Editor Report · Acceptance letter]

3 Feb 2022

PONE-D-21-29239R1 

Effects of *Catha Edulis* Forsk on Spatial Learning, Memory and Correlation with Serum Electrolytes in Wild- Type Male White Albino Rats 

Dear Dr. Limenie:

I'm pleased to inform you that your manuscript has been deemed suitable for publication in PLOS ONE. Congratulations! Your manuscript is now with our production department. 

Kind regards, 

on behalf of

Dr. Michelle Melgarejo da Rosa 

Academic Editor

PLOS ONE